# Atomic-Scale Friction on Monovacancy-Defective Graphene and Single-Layer Molybdenum-Disulfide by Numerical Analysis

**DOI:** 10.3390/nano10010087

**Published:** 2020-01-02

**Authors:** Haosheng Pang, Hongfa Wang, Minglin Li, Chenghui Gao

**Affiliations:** 1School of Mechanical Engineering and Automation, Fuzhou University, Fuzhou, Fujian 350002, China; 2Fujian Key Laboratory of Medical Instrumentation and Pharmaceutical Technology, Fuzhou University, Fuzhou, Fujian 350002, China

**Keywords:** numerical simulations, Prandtl–Tomlinson (PT) model, Schwoebel–Ehrlich barrier, monovacancy-defective graphene and single-layer molybdenum-disulfide (SLMoS_2_), atomic-scale friction

## Abstract

Using numerical simulations, we study the atomic-scale frictional behaviors of monovacancy-defective graphene and single-layer molybdenum-disulfide (SLMoS_2_) based on the classical Prandtl–Tomlinson (PT) model with a modified interaction potential considering the Schwoebel–Ehrlich barrier. Due to the presence of a monovacancy defect on the surface, the frictional forces were significantly enhanced. The effects of the PT model parameters on the frictional properties of monovacancy-defective graphene and SLMoS_2_ were analyzed, and it showed that the spring constant of the pulling spring *c_x_* is the most influential parameter on the stick–slip motion in the vicinity of the vacancy defect. Besides, monovacancy-defective SLMoS_2_ is found to be more sensitive to the stick–slip motion at the vacancy defect site than monovacancy-defective graphene, which can be attributed to the complicated three-layer-sandwiched atomic structure of SLMoS_2_. The result suggests that the soft tip with a small spring constant can be an ideal candidate for the observation of stick–slip behaviors of the monovacancy-defective surface. This study can fill the gap in atomic-scale friction experiments and molecular dynamics simulations of 2D materials with vacancy-related defects.

## 1. Introduction

Atomic-scale friction has been at the forefront of scientific interest over the decades. Generally, the study on the friction at atomic-scale is significant because it can result in a fundamental understanding of how friction happens as well as to facilitate the development of nanomechanical components [1,2,3]. With the advent of tip-based microscopy techniques such as the lateral force microscopy (LFM) [4], the study on atomic-scale frictional processes became accessible to researchers, and the field of nano-tribology has been set up since then [5,6,7,8]. Through the detection of torsional deflections of a cantilever as the tip is dragged over a surface, LFM can be applied to characterize the topographic and tribological features of nanomaterials [9,10]. The first observation of atomic-scale frictional phenomenon was reported by Mate et al. for a tungsten tip sliding on a graphite surface [4]. By means of a LFM, they found the atomic-scale stick–slip motion of the lateral force with the periodicity of graphite’s honeycomb structure. Since then, the origin and variation of atomic-scale friction have been thoroughly explored by tribologists.

As one of the most remarkable discoveries in nano-tribology, the atomic-scale stick–slip phenomenon appears in the time domain as a series of saw-tooth signals, and its period usually corresponds to the unit cell of the surface potential [11,12]. This observation can be theoretically reproduced within classical mechanics by using the Prandtl–Tomlinson (PT) model, which describes the movement of a point-like tip connected to a support by a harmonic spring in constant-force mode of an idealized LFM. As the support moves at a constant speed, the tip is dragged by the spring to slide over a whole surface, while at the same time, feeling the force from a corrugated tip–surface interaction potential featuring atomic periodicity [13,14,15,16]. The stick–slip instability occurs when the tip moves through the regions where the curvature of the tip–surface interaction potential exceeds the elastic constant of the pulling spring. Otherwise, such instabilities completely vanish, and the tip exhibits a continuous, low-dissipative motion, commonly known as superlubricity [17].

Atomically-thin laminar two-dimensional (2D) materials, such as graphene and single-layer molybdenum-disulfide (SLMoS_2_), are ideal candidates suited for atomic-scale frictional studies because these materials are air-stable, chemically inert [18], and can be easily cleaved to yield atomically flat surfaces with regular crystal structures [19]. However, the vacancy defects, known as the most commonly reported defects in graphene [20,21] and SLMoS_2_ [22,23], are inevitable during the production process and can result in considerable changes in surface morphology. So far, most experimental, theoretical, and simulation studies on the atomic-scale friction of graphene and SLMoS_2_ have focused on atomically flat surfaces [24,25,26,27,28]; nevertheless, these simplified idealizations cannot reflect the topography and frictional properties of many practical surfaces such as vacancy-defective surfaces. Molecular dynamics (MD) simulations have indicated that the frictional forces of defective graphene were significantly increased in the vicinity of vacancy point defects compared to that of the perfect graphene [29,30]. In the literature [29], Sun et al. thought the atomic friction of graphene with vacancy defects was similar to the sliding at an atomic-scale surface step and attributed the sharp variations in frictional forces to additional activation barriers (commonly referred to as the Schwoebel–Ehrlich barrier [31,32]) caused by the reduction of the atomic coordination at the vacancy defects. However, the sliding velocity of the tip in the above MD simulations was set to 2.5 m/s for [29] and 14.8 m/s for [30], respectively, which is much faster than that of typical experimental LFM set-ups (at the order of 10 nm/s [9]). Besides, the other parameters of the classical PT model, such as the effective masse of the system, the spring constant of the pulling spring, and the damping of the system, cannot be quantitatively adjusted in the MD simulation method or deduced from LFM experiments. Up to now, the effects of the PT model parameters on the atomic-scale frictional behaviors of the monovacancy-defective surface remain unclear, which hinders our understanding of the underlying mechanisms for atomic-scale frictional behaviors of defective 2D materials. As a supplement to the MD simulation and LFM experiments, the numerical approach is an efficient tool to obtain the theoretical solution of the PT model, recover the observed characteristics from the actual experiments and MD simulations, predict the results under conditions that cannot be tested experimentally and solved by MD simulations, and has been shown to accurately describe the frictional properties at atomic-scale surface steps [33,34].

In this work, we employ a novel numerical simulation method to explore the atomic-scale frictional properties of the monovacancy-defective graphene and SLMoS_2_. A modified interaction potential considering the Schwoebel–Ehrlich barrier is developed for the tip-defective surface interaction potential of the PT model. To study the mechanisms that control the frictional characteristics of the graphene and SLMoS_2_ with vacancy defects, we changed the effective masse of the system, the damping of the system, the sliding velocity of the tip, and the spring constant of the pulling spring to examine how the atomic-scale frictional behaviors depend on these variables. Ours is the first study revealing the influences of the classical PT model parameters on the atomic-scale frictional properties of a monovacancy-defective surface, and this work can make up for the deficiency of the LFM experiments and MD simulations when studying the atomic-scale frictional behaviors of 2D materials with vacancy-related defects.

## 2. Simulation Model and Method

Graphene consists of an isotropic hexagonal honeycomb lattice of carbon atoms. SLMoS_2_ consists of a 2D hexagonal honeycomb lattice where the Mo layer is covalently sandwiched between the bottom S layer and the top S layer. The vertical distance between two S layers of SLMoS_2_ is approximately 3.24 Å. The atomistic models of monovacancy-defective graphene and SLMoS_2_ are achieved by constructing a defect-free monolayer film first and then introducing the monovacancy defect in the center of the surface/top layer, as shown in the inserts in Figure 1a,b. In the adopted Cartesian coordinate system, the X-axis is set along the zigzag (ZZ) direction, the Y-axis along the armchair (AC) direction, and the Z-axis normal to the graphene and SLMoS_2_. The sliding was applied by moving the tip along the zigzag (ZZ) on the center line of the monovacancy-defective graphene and SLMoS_2_. The areas of the graphene and SLMoS_2_ film were 76.26 × 62.48 Å^2^ and 166.23 × 142.59 Å^2^ in the X–Y plane, respectively. The single-atom tip was generated for the atomic-scale friction according to the PT model, which consisted of a single carbon atom and was set as a rigid body in order to avoid the mass loss caused by the tip wear during the atomic-scale friction process. The constant-force mode is applied in the model and the loading force on the tip is kept at 2.5 nN. The schematic diagrams of the atomic-scale friction of monovacancy-defective graphene and SLMoS_2_ are shown in Figure 1a,b, respectively.

In the atomic-scale friction process, stick–slip behaviors can be explained by the classical PT model, which simplifies the single-asperity friction into one point-mass (tip) pulled along a corrugated potential by a driving support (spring) [16]. The classical PT model is shown as follows:(1)$$\begin{array}{l} m_{x}{\ddot{x}}_{t}=c_{x}\left(x_{M}-x_{t}\right)-\frac{\partial V_{\mathrm{int}}\left(x_{t},y_{t}\right)}{\partial x_{t}}-\gamma_{x}{\dot{x}}_{t}\\ m_{y}{\ddot{y}}_{t}=c_{y}\left(y_{M}-y_{t}\right)-\frac{\partial V_{\mathrm{int}}\left(x_{t},y_{t}\right)}{\partial y_{t}}-\gamma_{x}{\dot{y}}_{t} \end{array}$$
where *c_x_* and *c_y_* represent the spring constants of the pulling spring, *m_x_* and *m_y_* represent the effective masses of the system, *x_t_* and *y_t_* represent the actual positions of the tip, *x_M_* = *y_M_* = *V_m_t* represent the equilibrium positions of the tip without potential, *V_m_* represents the sliding velocity of the tip, *V*_int_ (*x_t_*, *y_t_*) represents the tip–surface interaction potential, *γ_x_* and *γ_y_* represent the damping of the system. The frictional forces *F_x_* = *c_x_* (*x_M_* − *x_t_*) and *F_y_* = *c_y_* (*y_M_* − *y_t_*) of the PT model are composed of a dissipative part and a conservative part, which are given by the damping term and the tip–surface interaction potential, respectively [15]. Theoretically, the degree of freedom for the atomic motions is high in 2D systems where the non-adiabatic motion is avoided [28]. In the PT model, the sweeping of the tip can be regarded as adiabatic.

The tip–surface interaction potential of the classical PT model in Equation (1) is governed by the corrugated potential *V_int_* (*x_t_*, *y_t_)*, which can be described as
(2)$$V_{\mathrm{int}}\left(x_{t},y_{t}\right)=V_{0}\cos\left(\frac{2\pi }{a_{x}}x_{t}\right)\cos\left(\frac{2\pi }{a_{y}}y_{t}\right)$$
where *V_0_* represents the initial value of tip-sample interaction, *a_x_* and *a_y_* represent the sample’s lattice constants in the ZZ and AC orientation, respectively. In the ZZ orientation, *a_x_* is 2.46 and 3.16 Å for graphene and SLMoS_2_, respectively.

In order to quantify the tip–surface interaction potential, the Lennard–Jones (LJ) potential *V*_LJ_, which has been used to accurately describe the frictional properties at atomic-scale steps [33,34], is taken into consideration as the tip–surface potential in the case of the single-atom tip crossing the monovacancy defects of graphene and SLMoS_2_. The *V*_LJ_ can be described as follows:(3)$$V_{\mathrm{LJ}}=\sum {}_{i=1}^{N}E_{0}\left({\left(r_{0}/r_{i}\right)}^{12}-2{\left(r_{0}/r_{i}\right)}^{6}\right)$$
where *r_i_* represents the distance between the *i*th surface atom and the tip, *r*_0_ and *E*_0_ represent the equilibrium distance and the binding energy, respectively.

In the presence of the vacancy defects, the surface state is similar to that of the atomic-scale surface steps. According to previous LFM studies on the monovacancy-defective graphene [29,30] and the stepped surfaces [34,35,36,37,38], there is the Schwoebel–Ehrlich barrier existing in the vicinity of the vacancy defect, and hence the long-range interactions *V*_long_ are varying, which means a second contribution to the tip–surface interaction potential should be considered. A modified interaction potential *V*_total_ containing the periodic interaction *V*_int_ (*x_t_*, *y_t_*) between the tip and surface as well as the sharply increasing potential barrier at the vacancy defect site simulating the long-range interaction potential *V*_long_ is constructed to reflect the tip–surface interaction of the defective graphene and SLMoS_2_ with monovacancy defects, which is shown as follows:(4)$$V_{\mathrm{total}}=V_{\mathrm{int}}(x_{t},y_{t})+V_{\mathrm{long}}$$
The long-range interaction potential *V*_long_ can be described by the model [33,37,38] based on the approximate low-load analytical form of the Schwoebel–Ehrlich barrier in the vicinity of the atomic-scale surface step. The *V*_long_ is shown as follows:(5)$$V_{\mathrm{long}}\left(x\right)=E\left[-\mathrm{erf}\left(\frac{x}{b_{1}}-d\right)+\mathrm{erf}\left(\frac{x-c}{b_{2}}\right)\right]$$
where *E* represents a constant of the order of an electronvolt, *b*_1_ represents the effective barrier width at the vacancy defect site, *b*_2_, *c,* and *d* are the constants representing a recovery of the potential away from the vacancy defect.

In order to obtain the numerical description of the frictional force *F_x_* in the PT model, we develop the modified interaction potential *V*_total_ by fitting the LJ potential curve and set it as the approximate tip–surface interaction potential of the monovacancy-defective graphene and SLMoS_2_. Relevant parameters for the LJ potential *V*_LJ_ and the long-range interaction *V*_long_ are shown in Appendix A, respectively. The simulations of the atomic-scale frictional behaviors in our work are performed by using the software Mathematica (Wolfram Research, Champaign, IL, USA, version 12).

## 3. Results and Discussions

We first calculate the tip–surface interaction potential by LJ potential (*V*_LJ_) and then develop the modified interaction potential (*V*_total_) through fitting the LJ potential (*V*_LJ_). By introducing the potential *V*_tota_ into Equation (1), we can calculate the path of the tip on the sample surface and obtain the frictional forces. Finally, we discuss the effects of the parameters (the effective mass of the system *m*, damping of the system *γ*, sliding velocity *V_m_*, and spring constant of the pulling spring *c*) of the classical PT model on the frictional force and the stick-slip behaviors of monovacancy-defective graphene and SLMoS_2_.

Figure 2a,b shows the tip–surface potential *P_t_* versus the sliding distance *D* curves when the tip moves over the vacancy defect at the surface of the defective graphene and SLMoS_2_, respectively. The corresponding frictional force *F_x_* versus the sliding distance *D* curves are shown in Figure 2c,d, respectively. The typical parameters of LFM experiments [15,28,39] are set as follows: *m_x_* = *m_y_* = 10^−8^ kg, *c_x_* = *c_y_* = 10 N/m, *V*_0_ = 1.0 eV, and *V*_m_ = 40 nm/s. The damping of the system can be obtained with *γ_x_ = γ_y_ =* 2 (*c_x_ m_x_*)^1/2^ ≈ 10^−3^ N·s/m. Owing to the presence of the monovacancy defect, the changes in the tip–surface potential and frictional force of the monovacancy-defective graphene and SLMoS_2_ can be clearly observed. There are the large positive tip–surface potential and frictional force occurring when the tip approaches the vacancy, indicating the attraction from the vacancy. After passing the vacancy, the tip feels a large attractive force from the vacancy again. The enhanced frictional force in the vicinity of the vacancy defect is in agreement with that in the previous works about the monovacancy-defective graphene [29,30], and the sharp variations in tip–surface potentials and frictional forces can be attributed to the Schwoebel–Ehrlich barriers [31,32], which have been already observed in the friction at the atomic-scale surface steps of graphene [34,35,36,38] and MoS_2_ [37].

In order to explore the stick–slip motions during the frictional process of the monovacancy-defective graphene and SLMoS_2_, we plot the position of the single-atom tip versus the sliding distance curves, as illustrated in Figure 2e,f. It indicates that the stick–slip motion is absent for the friction of the monovacancy-defective graphene, while the obvious stick–slip motion appears in the vicinity of vacancy defect of monovacancy-defective SLMoS_2_. It can be attributed to the difference in the curvature of the tip–surface interaction *V*″ (or the critical elastic constant *c_x_*) at the vacancy defect site of defective graphene and SLMoS_2_ (*V*″_carbon_ = 0.91 for the monovacancy-defective graphene and *V*″_SLMoS2_ = 17.18 for the monovacancy-defective SLMoS_2_) because the stick–slip phenomenon occurs on condition that the elastic constant *c_x_* of the pulling spring (*c_x_* = 10 N/m, *c_x_* > *V*″_carbon_, and *c_x_* < *V*″_SLMoS2_) should be less than or equal to the curvature of the tip–surface interaction potential *V*″ [17,34]. Besides, although there is no stick–slip motion for the monovacancy-defective graphene, the single-atom tip does undergo periodical acceleration–deceleration motion without a full stick, which is similar to the stick–slip motion, and in accord with previous studies of defective graphene [30].

### 3.1. Variation of the Effective Masses m_x_ of the System

The relations of the frictional force *F_x_* vs. the sliding distance *D* for four effective masses *m_x_* of the system (namely, *m_x_* = 10^−10^, 5 × 10^−9^, 3 × 10^−8^, and 12 × 10^−8^ kg) are shown in Figure 3a,b. The *c**_x_*, *V*_m_*,* and *γ_x_* are set to 10 N/m, 40 nm/s, and 10^−3^ N·s/m, respectively. It is seen that all the *F_x_*-*D* curves of the monovacancy-defective graphene are almost the same, as shown in Figure 3a. For the monovacancy-defective SLMoS_2_, however, the amplitude of the minimum frictional force *F_x_*_min_ increases with *m_x_* (namely, the amplitude of *F_xmin_* is 0.465, 0.468, 0.605, and 1.040 nN for *m_x_* = 10^−10^, 5 × 10^−9^, 3 × 10^−8^, and 12 × 10^−8^ kg, respectively), and the other frictional forces including the maximum frictional force *F_x_*_max_ and the periodic frictional forces (namely, the frictional forces except for that in the vicinity of the vacancy defect) remain basically unchanged, as shown in Figure 3b.

The relations of the position of the single-atom tip *P* vs. the sliding distance *D* for the above four effective masses *m_x_* of the system are shown in Figure 3c,d. It can be seen that there is virtually no change in the *P* vs. *D* curves of the monovacancy-defective graphene and SLMoS_2_ with different *m_x_*, which means the stick–slip motion in the vicinity of the vacancy defect is nearly independent of the effective masses *m_x_* of the system.

### 3.2. Variation of the Damping γ_x_ of the System

The relations of the frictional force *F_x_* vs. the sliding distance *D* for four different dampings *γ_x_* of the system (namely, *γ_x_* = 8 × 10^−5^, 4 × 10^−4^, 2 × 10^−3^, and 10^−2^ N·s/m) are shown in Figure 4a,b. The *c_x_*, *V*_m_*,* and *m_x_* are set to 10 N/m, 40 nm/s, and 10^−8^ kg, respectively. The results show that the frictional forces *F_x_* of the monovacancy-defective graphene and SLMoS_2_ increase with the damping *γ_x_*. It can be explained by Equation (1) that the frictional force *F_x_* = *c_x_* (*x_M_* − *x_t_*) = $m_{x}{\ddot{x}}_{t}+\partial V_{\mathrm{int}}\left(x_{t},y_{t}\right)/\partial x+\gamma_{x}{\dot{x}}_{t}$ is positively correlated with the damping *γ_x_*, which is in agreement with the definition of the frictional force in the PT model [15]. However, the amplitudes of the frictional forces including the enhanced frictional forces in the vicinity of the vacancy defect and the periodic frictional forces of the monovacancy-defective graphene and SLMoS_2_ increase with decreasing *γ_x_*, as shown in Appendix A. This negative correlation between the amplitude of the frictional force and the damping *γ_x_* is in accord with the physical meaning of the damping term in the PT model, namely, the term considering the mechanisms of the energy dissipation in atomic-scale friction [15,16].

The relations of the position of the single-atom tip *P* vs. the sliding distance *D* for the above four dampings *γ_x_* of the system are shown in Figure 4c,d. The results show that the *P* vs. *D* curves of the monovacancy-defective graphene with different *γ_x_* are coincident, and this trend is similar to that of the monovacancy-defective SLMoS_2_ except for a little shift in the stick–slip regime at the defect position for the case of *γ_x_* = 10^−2^ N·s/m. It indicates that the damping *γ_x_* of the system has little effect on the stick–slip motions in the vicinity of the vacancy defect.

### 3.3. Variation of the Sliding Velocity V_m_

Figure 5a,b shows the relations of the frictional force *F_x_* versus the sliding distance for five different sliding velocities *V*_m_ (namely, *V*_m_ = 2, 15, 60, and 150 nm/s). The *γ_x_*, *c_x_,* and *m_x_* are set to 10^−3^ N·s/m, 10 N/m, and 10^−8^ kg, respectively. It is seen that the frictional forces of the monovacancy-defective graphene and SLMoS_2_ increase with the *V_m_*. The result can be explained by Equation (1) that the atomic-scale frictional force *F_x_* = *c_x_* (*x_M_* − *x*_t_) = *c_x_* (*V*_m_*t* − *x*_t_) [15,34] is positively correlated with the sliding velocity *V*_m_, which has been demonstrated in previous work [40]. Besides, the amplitudes of the frictional forces including the enhanced frictional forces in the vicinity of the vacancy defect and the periodic frictional forces of the monovacancy-defective graphene and SLMoS_2_ almost keep constant at different sliding velocities *V*_m_.

The relations of the position of the single-atom tip *P* vs. the sliding distance *D* for the above four sliding velocities *V*_m_ are shown in Figure 5c,d. Similar to the trends of *m_x_* and *γ_x_*, the variations in the *P* vs. *D* curves of the monovacancy-defective graphene and SLMoS_2_ with different *V*_m_ are extremely small and can be neglected, which indicates that the stick–slip motion in the vicinity of the vacancy defect is approximately independent of the sliding velocity *V*_m_. The result seems to contradict previous studies suggesting that stick–slip motions can also be suppressed or even ruined by thermally activated jumps when the tip scans at a low velocity [41,42]. Nevertheless, this is because the temperature was not considered for the classical PT model in this work, and hence the resulting thermally activated fluctuations of the friction behaviors were not observed. We expect the effect of the temperature on the atomic-scale frictional properties of defective 2D materials to be systematically studied in future works.

### 3.4. Variation of the Spring Constants c_x_ of the Pulling Spring

Due to the great difference in the critical spring constants *c_x_* (the curvature of the tip–surface interaction potential) of the monovacancy-defective graphene and SLMoS_2_ in this work (the critical *c_x_* is 0.91 and 17.18 N/m for the monovacancy-defective graphene and SLMoS_2_, respectively, as calculated in the beginning of Section 3), it is reasonable to set three different *c_x_* which are based on their respective critical values *c_x_* and separated by a amplitude of 5 times. Specifically, the small spring constants *c_x_*, the critical spring constants *c_x_*, and the larger spring constants *c_x_* of the monovacancy-defective graphene are set to 0.18, 0.91, and 4.55 N/m, respectively. The small spring constants *c_x_*, the critical spring constants *c_x_*, and the larger spring constants *c_x_* of the monovacancy-defective SLMoS_2_ are set to 3.44, 17.18, and 85.9 N/m, respectively. Figure 6a,b show the relations of the frictional force *F_x_* versus the sliding distance for the three different spring constants *c_x_* of the pulling spring. The *γ_x_*, *V*_m_*,* and *m_x_* are set to 10^−3^ N·s/m, 40 nm/s, and 10^−8^ kg, respectively. It is seen that the amplitudes of the frictional forces of the monovacancy-defective graphene and SLMoS_2_ increase with the spring constants *c*. However, the details of the observed variations in the frictional forces of two materials are distinctly different. For the monovacancy-defective graphene, all the frictional forces including the enhanced frictional forces in the vicinity of the vacancy defect and the periodic frictional forces are positively correlated with the spring constants *c_x_*. For the monovacancy-defective SLMoS_2_, the maximum frictional force *F_x_*_max_ and the periodic frictional forces almost keep constant at different *c_x_* while the amplitude of the minimum frictional force *F_x_*_min_ in the vicinity of vacancy point defects increases with the *c_x_*.

The relations of the position of the single-atom tip *P* vs. the sliding distance *D* for the above three spring constants *c_x_* are shown in Figure 6c,d. It is observed that when the *c_x_* of the monovacancy-defective graphene and SLMoS_2_ are less than or equal to the critical value, the stick–slip behaviors occur and become more obvious as the spring constants *c_x_* decrease (the black and red wavy lines in Figure 6c,d). Otherwise, the stick–slip behaviors disappear completely (the blue lines in Figure 6c,d). The results are in agreement with previous studies [15,17,34,43]. In addition, it is noted that the stick–slip behaviors in the vicinity of the vacancy defect of the monovacancy-defective SLMoS_2_ are much more obvious than that of the monovacancy-defective graphene when their respective spring constants *c_x_* are equal to or a fifth of the critical *c_x_* (see the dotted boxes in Figure 6c,d). It indicates that it is more sensitive to the stick–slip motion at the vacancy defect site for the monovacancy-defective SLMoS_2_ than that for the monovacancy-defective graphene, which can be due to the more complicated three-layer-sandwiched atomic structure of SLMoS_2_. It is worth noting that SLMoS_2_ [22] can be charged, which may influence the stick–slip motions around the vacancy defect. However, the interaction between the LFM tip and the SLMoS_2_ with the charged state of the vacancy defect remains unclear so far, and we expect this issue will be deeply studied in future works. Besides, the influence of the chirality on the atomic-scale friction of monovacancy-defective 2D graphene and SLMoS_2_ is also explored, and the results suggest that the variation trends of the stick–slip behaviors and the frictional forces along the AC orientation are similar to that along the ZZ orientation, as shown in Appendix A. Compared with the other parameters of the classical PT model (the effective masse of the system *m_x_*, damping of the system *γ_x_*, and sliding velocity *V_m_*), the spring constant of the pulling spring *c_x_* has the most remarkable impact on the stick–slip motion in the vicinity of the vacancy defect, which suggests that the soft tip with small spring constant can be suitable for the observation of the stick–slip motions in the atomic-scale friction experiments of the monovacancy-defective 2D materials. Otherwise, the stiff tip with larger spring constant can be suitable for the observation of the low-dissipative motions or the superlubricity phenomena in the atomic-scale friction experiments of the monovacancy-defective 2D materials.

## 4. Conclusions

In this work, we employed a numerical simulation method to explore the atomic-scale friction of monovacancy-defective 2D graphene and SLMoS_2_ based on a classical PT model with a modified interaction potential considering the Schwoebel–Ehrlich barrier. It was found that the frictional forces are significantly enhanced due to the presence of a monovacancy defect on the surface. The effects of the PT model parameters (the effective masse of the system *m_x_*, damping of the system *γ_x_*, sliding velocity *V*_m_, and spring constant of the pulling spring *c_x_*) on the frictional properties of monovacancy-defective graphene and SLMoS_2_ were analyzed. It revealed that the spring constant of the pulling spring *c_x_* has the most remarkable impact on the stick–slip motion in the vicinity of the vacancy defect compared with other parameters. In addition, it is more sensitive to the stick–slip motion at the vacancy defect site for the monovacancy-defective SLMoS_2_ than that for the monovacancy-defective graphene, which can be due to the complicated three-layer-sandwiched atomic structure of SLMoS_2_. This study suggests that a soft tip with small spring constant can be an ideal candicate for the LFM experiment of stick–slip behaviors of the monovacancy-defective surface, and it can provide valuable complementary information for atomic-scale friction experiments and MD simulations of 2D materials with vacancy-related defects.

## Figures and Tables

**Figure 1 nanomaterials-10-00087-f001:**
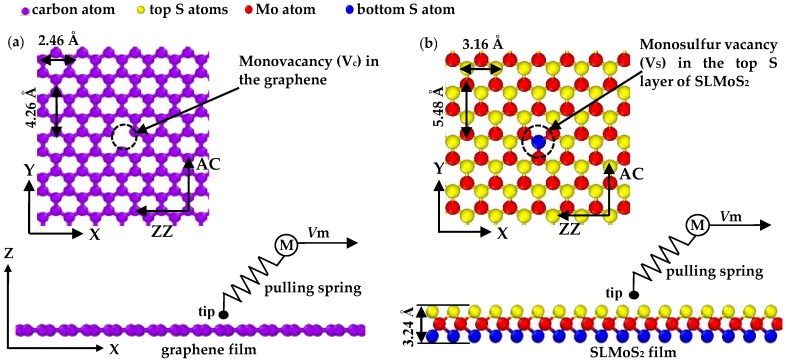
(**a**) The schematic diagram of the atomic-scale: the frictional interaction between a single-atom tip and the graphene with the monovacancy (Vc). (**b**) The schematic diagram of the atomic-scale: the frictional interaction between a single-atom tip and the SLMoS_2_ with the monosulfur vacancy (Vs) in the top S layers. The tip is connected via a pulling spring with the elasticity of C to the body M and is moved along the X direction at a constant velocity of *V_m_*.

**Figure 2 nanomaterials-10-00087-f002:**
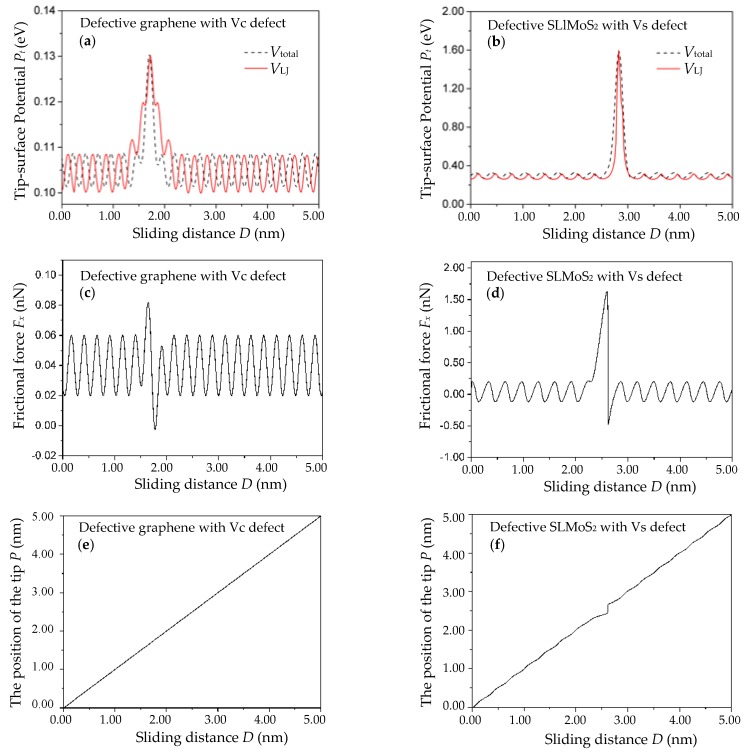
The tip-surface potential *Pt* vs. scanning distance *D* curves (**a**,**b**), the frictional force *F_x_* vs. the sliding distance *D* curves (**c**,**d**), and the position of the single-atom tip *P* vs. the sliding distance *D* curves (**e**,**f**) of the monovacancy-defective graphene and SLMoS_2_. The solid red curves and the dashed black curves in (**a**,**b**) were obtained by calculating the LJ potential (*V*_LJ_) and by fitting the LJ potential (*V*_LJ_) via the modified interaction potential (*V*_total_), respectively.

**Figure 3 nanomaterials-10-00087-f003:**
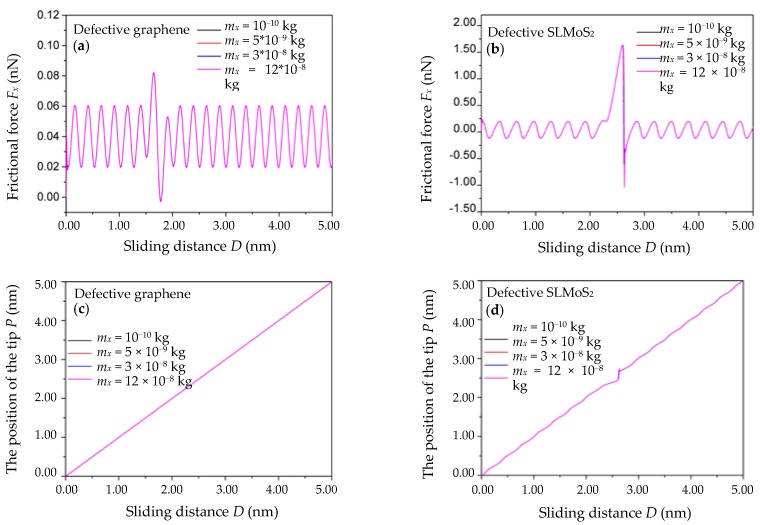
The frictional force *F_x_* vs. the sliding distance *D* curves of the monovacancy-defective graphene (**a**) and SLMoS_2_ (**b**) with different effective masses *m_x_* of the system. The position of the single-atom tip *P* vs. the scanning distance *D* curves of the monovacancy-defective graphene (**c**) and SLMoS_2_ (**d**) with different effective masses *m_x_* of the system. The curves in Figure 3 look like a single curve, but they are the overlap of multiple curves, and the displayed curves are the pink curves.

**Figure 4 nanomaterials-10-00087-f004:**
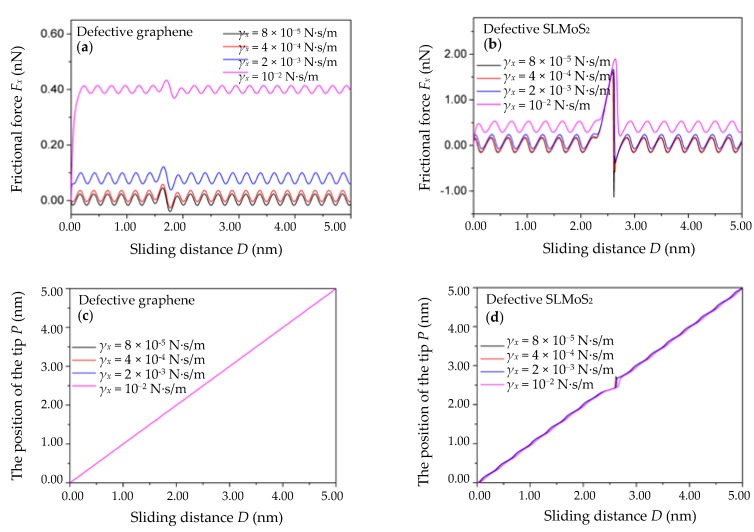
The frictional force *F_x_* vs. the sliding distance *D* curves of the monovacancy-defective graphene (**a**) and SLMoS_2_ (**b**) with different dampings *γ_x_* of the system. The position of the single-atom tip *P* vs. the scanning distance *D* curves of the monovacancy-defective graphene (**c**) and SLMoS_2_ (**d**) with different dampings *γ_x_* of the system. The curves in Figure 4 look like a single curve, but they are the overlap of multiple curves, and the displayed curves are the pink curves.

**Figure 5 nanomaterials-10-00087-f005:**
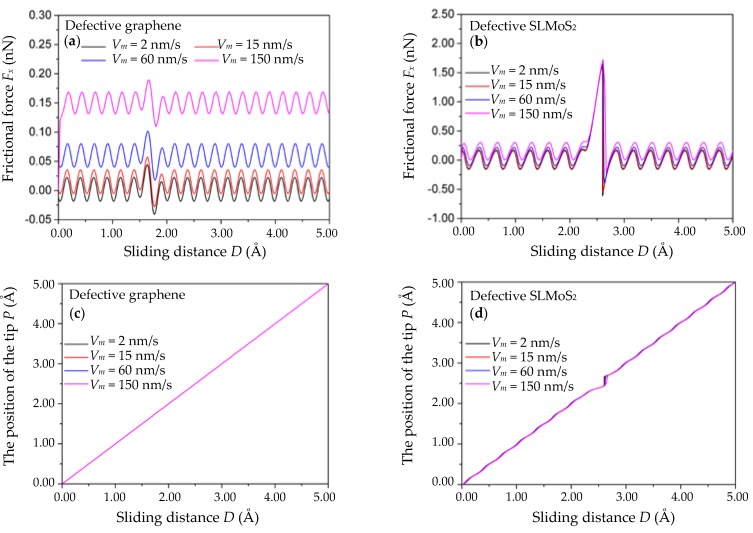
The frictional force *F_x_* vs. the sliding distance *D* curves of the monovacancy-defective graphene (**a**) and SLMoS_2_ (**b**) with different sliding velocities *V*_m_. The position of the single-atom tip *P* vs. the scanning distance *D* curves of the monovacancy-defective graphene (**c**) and SLMoS_2_ (**d**) with different sliding velocities *V*_m_. The curves in Figure 5 look like a single curve, but they are the overlap of multiple curves, and the displayed curves are the pink curves.

**Figure 6 nanomaterials-10-00087-f006:**
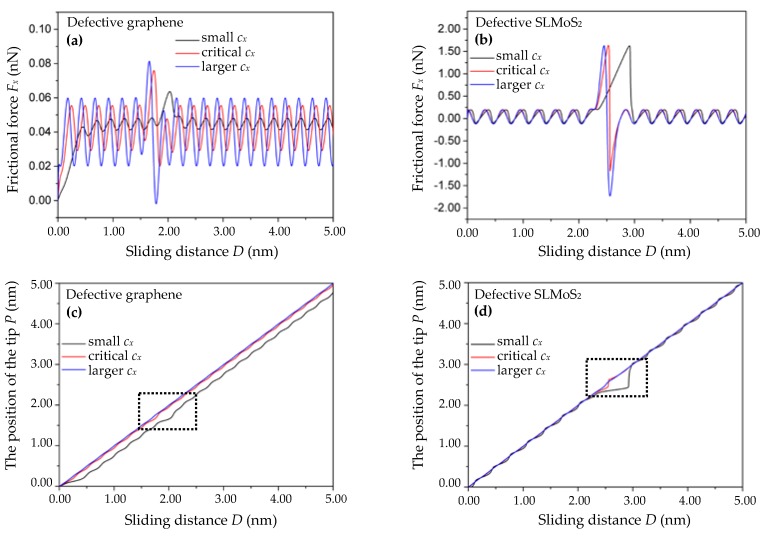
The frictional force *F_x_* vs. the sliding distance *D* curves of the monovacancy-defective graphene (**a**) and SLMoS_2_ (**b**) with different spring constants *c_x_* of the pulling spring. The position of the single-atom tip *P* vs. the scanning distance *D* curves of the monovacancy-defective graphene (**c**) and SLMoS_2_ (**d**) with different *c_x_*. The small, critical, and larger *c_x_* of the monovacancy-defective graphene in Figure 6a is 0.18, 0.91, and 4.55 N/m, respectively. The small, critical, and larger *c_x_* of the monovacancy-defective SLMoS_2_ in Figure 6b is 3.44, 17.18, and 85.9 N/m, respectively.

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
