# Peer review of "Atomic-Scale Friction on Monovacancy-Defective Graphene and Single-Layer Molybdenum-Disulfide by Numerical Analysis"

_nanomaterials, 2020, doi:10.3390/nano10010087_

Round 1

Reviewer 1 Report

Authors have shown results of numerical simulations for atomic-scale friction of single layer graphene and MoS2 with mono-vacancy. One of valuable information from this work is that a soft tip would be a good candidate for the LFM experiments. This would be very helpful for the experimental scientist. I suggest that this work can be published after reflecting comments as follows:

‘Numerical Analysis’ has to be included into the ‘Title ‘. For the LFM experiments, the atomically thick materials have to be placed onto a substrate. It is known that an underlying substrate of atomically thick materials is a major contributor to the friction. Can authors discuss on the effect of the underlying substrate on the friction of graphene and MoS2 with mono-vacancy? As mentioned, one of valuable information from this work is that a soft tip would be a good candidate for the LFM experiments. Is there any experimental observation supporting this conclusion? For example, a relationship between spring constant of AFM probe and stick-slip friction.

Author Response

Response:

Dear reviewer, thank you for your valuable suggestions, and some modifications have been made to the paper

Point 1: ‘Numerical Analysis’ has to be included into the ‘Title‘.

Response 1. We have change the ‘Title‘ as ‘Atomic-scale friction on monovacancy-defective graphene and single-layer molybdenum-disulfide by numerical analysis‘ in lines 3-4.

Point 2: For the LFM experiments, the atomically thick materials have to be placed onto a substrate. It is known that an underlying substrate of atomically thick materials is a major contributor to the friction. Can authors discuss on the effect of the underlying substrate on the friction of graphene and MoS2 with mono-vacancy?

Response 2. This suggestion is very good. There are two situations for the cases considering the substrate: (1) if the specimen is fixed on the substrate, the interaction force between the tip and the substrate depends on the cutoff radius of LJ potential. Generally, the cutoff radius is not set too larger, so the substrate is out of the interaction of LJ potential. In this situation, the presence or absence of a substrate has little effect on the atomic-scale friction

(2) if the specimen is freedom or can be moved on the substrate, so the position and velocity of every atom of the specimen require to be calculated in our work. It is difficult to realize by the numerical simulation method, and the molecular dynamics simulation can be the suitable method for this situation. However, the advantages of the numerical simulation method cannot work.

Point 3: As mentioned, one of valuable information from this work is that a soft tip would be a good candidate for the LFM experiments. Is there any experimental observation supporting this conclusion? For example, a relationship between spring constant of AFM probe and stick-slip friction. 

Response 3. One of the conclusions from our work is that a soft tip would be a good candidate for the LFM experiments. The experimental observation supporting this conclusion can be found in ref 17 in the “Introduction” (lines 52) and “Result and discussion” (lines 308). In ref. 17, a transition from stick-slip to continuous sliding (see the Figure 1 (a), (b) and (c) in ref. 17) is observed for atomically modulated friction on the atomically flat surface of NaCl single crystals by the FFM (friction force microscope) experiments. When the stick-slip instabilities cease to exist, a new regime of ultralow friction is encountered. The transition is described in the framework of the Tomlinson model using a parameter η which relates the strength of the lateral atomic surface potential and the stiffness of the contact under study. Hence we think the relationship between spring constant of AFM probe and stick-slip friction has been verified by the experimental work in ref. 17. To the best of our knowledge, there are no related experimental works on the monovacancy-defective 2D materials, ours is the first work on theoretical aspects, and we hope it can provide valuable reference for the future experimental works on the monovacancy-defective 2D materials. As the supplement, we have added it “Otherwise, the stiff tip with larger spring constant can be suitable for the observation of the low-dissipative motions or the superlubricity phenomena in the atomic-scale friction experiments of the monovacancy-defective 2D materials.” in lines 326-329.

Ref. [17] in the manuscript: A. Socoliuc; R. Bennewitz; E. Gnecco; Meyer, E. Transition from stick-slip to continuous sliding in atomic friction: Entering a new regime of ultralow friction. Phys. Rev. Lett. 2004, 92, 134301.

Reviewer 2 Report

Manuscript ID:   nanomaterials-675110

Title:                     Atomic-scale friction on monovacancy-defective graphene and single-layer molybdenum-disulfide 

Authors:                        Haosheng Pang, Hongfa Wang, Minglin Li, Chenghui Gao

Dear authors,

The influence of lattice defect at uppermost surface of tribo-materials is one of topic in lubrication. This work challenges to understand the phenomena by simulation. Following comments are addressed.

1. Figure 1 models Schottky defects. Frenkel defect could occur under tribological conditions. Are there any differences between these crystal defects? The balance of electric charge around the defect should be considered for these models. Graphene could be neutral but molybdenum disulfide should be charged. It can influence the stick-slip motion shown in Figures 2(f), 3(d), 4(d), and 5(d).

2. Use SI unit instead of Angstrom.

3. Tip material should be considered.

4. Direction of sliding with regard to lateral structure should be considered.

Author Response

Response:

Dear reviewer, thank you for your valuable suggestions, and some modifications have been made to the paper.

Point 1: Figure 1 models Schottky defects. Frenkel defect could occur under tribological conditions. Are there any differences between these crystal defects? The balance of electric charge around the defect should be considered for these models. Graphene could be neutral but molybdenum disulfide should be charged. It can influence the stick-slip motion shown in Figures 2(f), 3(d), 4(d), and 5(d).

Response 1. This suggestion is very good. The Schottky defect is defined as the vacancy defect in specimen’s the surface, and the total number of all the atoms in the specimen containing the Schottky defect is reduced due to the presence of the Schottky defect. The Frenkel defect is defined as the vacancy- interstitial pair defect in the specimen’s surface. Specifically, an atom (or ion) that has occupied a lattice previously leaves the original lattice and becomes a interstitial atom (or ion), resulting in a vacancy at the lattice where it occupied previously. The total number of all the atoms in the specimen with the Frenkel defect remains unchanged because there is no loss of surface atoms, and the vacancy atom (or ion) just move to another location in the specimen’s surface. Therefore, the Schottky defect is different from the Frenkel defect.

We agree the review’s suggestion on the electric charge around the defect. However, we don't know the specific distribution of the electric charge. Although the latest work by first-principles calculations (Arunima Singh*, Pooja Basera, Shikha Saini, Manish Kumar and Saswata Bhattacharya. “Importance of Many-Body Dispersion in the Stability of Vacancies and Antisites in Free-Standing Monolayer of MoS2 From First-Principles Approaches”, The Journal of Physical Chemistry, to be published) have found that the single-layer MoS2 with +2 charged state of the the vacancy defect is stable, nevertheless, the interaction between the LFM tip and the SLMoS2 with the charged state of the vacancy defect remains unclear so far. Hence we think it is a challenging work, and will explore it in the future. We have added the sequence “It is worth noting that SLMoS2 [22] can be charged, which may influence the stick-slip motions around the vacancy defect. However, the interaction between the LFM tip and the SLMoS2 with the charged state of the vacancy defect remains unclear so far, and we expect this issue can be deeply studied in the future works.” in lines 314-317. The references [20] and [22] are the same in the original manuscript.

Point 2. Use SI unit instead of Angstrom.

Response 2. According to the reviewer’s suggestions, we have used SI unit in the whole paper including all the texts and figures.

Comment 3. Tip material should be considered.

Response 3. According to the reviewer’s suggestions, we use the single carbon atom as the single-atom tip and set it as a rigid body in order to avoid the mass loss caused by the tip wear during the atomic-scale friction process. We have added the sequence “, which consisted of single carbon atom and was set as a rigid body in order to avoid the mass loss caused by the tip wear during the atomic-scale friction process” in lines 104-106 and changed the sequence “T means the carbon atom of the single-atom tip” in the note of Table 1 in Supporting Materials.

Point 4. Direction of sliding with regard to lateral structure should be considered.

Response 4. According to the reviewer’s suggestions, we considered the direction of sliding with regard to lateral structure by adding the Figure S2 and S3 in the supporting materials, and add the sentence “Besides, the influence of the chirality on the atomic-scale friction of monovacancy-defective 2D graphene and SLMoS2 is also explored, and the results suggest that the variation trends of the stick-slip behaviors and the frictional forces along AC orientation are similar to that along ZZ orientation, as shown in Figure S2 and S3 in the supporting materials.” in lines 317-321.

Round 2

Reviewer 2 Report

Manuscript ID:    nanomaterials-675110

Title:                                 Atomic-scale friction on monovacancy-defective graphene and single-layer molybdenum-disulfide

Authors:                            Haosheng Pang, Hongfa Wang, Minglin Li, Chenghui Gao

Dear Authors,

Thank you for the updated manuscript. Now it is acceptable.